# Factors associated with hypotension during the first hour of continuous renal replacement therapy in critically Ill patients: A prospective observational study

Mengdie Xue[1], Suying Lu[2], Chenglin Zhao[2], Zheyao Zhang[3], Jing Yang[2], Zhiyu Mao[1], Jingjuan Xu [2]*

1 School of Nursing, Suzhou Medical College of Soochow University, Suzhou, China, 2 The Third Affiliated Hospital of Soochow University, Changzhou, China, 3 School of Medical Imaging, Xuzhou Medical University, Xuzhou, China

* heleneshue@sina.com

## Abstract

### Purpose

To investigate the factors of hemodynamic instability within the first hour of continuous renal replacement therapy (CRRT) in critically ill patients.

### Materials and methods

A prospective observational cohort study of patients admitted to the intensive care unit (ICU) and underwent CRRT between January 17, 2024, and December 25, 2024, was conducted. The least absolute shrinkage and selection operator (LASSO) regression was used to screen potential factors and a multivariate logistic regression model was performed to determine the independent factors of hypotension within the first hour of CRRT.

### Results

Hypotension occurred in 166 out of 435 patients (38.2%). Female (*OR*=0.53, 95%*CI*:0.3–0.89), the use of colloidal solutions (*OR*=0.23, 95%*CI*:0.12–0.46), platelet count (PLT) (*OR*=0.99, 95%*CI*:0.99–0.99), and baseline mean arterial pressure (MAP) (*OR*=0.95, 95%CI:0.93–0.96) were recognized as protective factors against hypotension within the first hour of CRRT in critically ill patients. Older age (*OR*=1.02, 95%*CI*:1.01–1.04), mechanical ventilation (*OR*=2.59, 95%*CI*:1.28–5.23), ultrafiltration rates of 101-200 mL/h (*OR*=2.04, 95%*CI*:1.11–3.75), international normalized ratio (INR) (*OR*=1.78, 95%*CI*:1.02–3.09), and high myoglobin level (*OR*=1.01, 95%*CI*:1.01–1.01) were identified as significant risk factors. Baseline MAP and mechanical ventilation are the most important predictors of hypotension within the first hour of CRRT in critically ill patients.

**Data availability statement:** The minimal anonymized dataset necessary to replicate the study findings has been uploaded to Zenodo under the following DOI: 10.5281/zenodo.15259287.

**Funding:** The author(s) received no specific funding for this work.

**Competing interests:** The authors have declared that no competing interets exist.

## Conclusions

The incidence of hypotension within the first hour of CRRT in critically ill patients was 38.2%. Female, PLT, baseline MAP are protective factors, and age, mechanical ventilation, ultrafiltration rate of 101-200ml/h, INR, myoglobin are risk factors for hypotension within the first hour of CRRT in critically ill patients.

## Introduction

Continuous Renal Replacement Therapy (CRRT) is a set of therapies that continuously and gradually remove solutes and excess fluids from the body [1]. It plays a vital role in providing multiorgan support and is the primary modality of Renal Replacement Therapy (RRT) for critically ill patients [2,3]. Compared to other dialysis techniques, CRRT exhibits superior hemodynamic stability [4,5]. Nonetheless, the incidence of intra-dialytic hypotension (IDH) during CRRT remains high, ranges from 43% to 87% [6].

IDH during CRRT is a critical issue affecting patients' outcome. Studies have shown that the incidence of IDH is highest within the first hour of CRRT, with approximately two-thirds of patients experiencing IDH during this period [7–9]. The occurrence of IDH not only compromises dialysis adequacy and therapeutic efficacy but also worsens patients' prognosis. When IDH occurs, clinicians often need to reduce the ultrafiltration rate or even temporarily stop treatment, which compromises solute clearance and fluid management. In addition, recurrent hypotensive episodes can lead to prolonged dependence on vasopressors, increased need for mechanical ventilation, and extended ICU length of stay. IDH is independently linked to in-hospital mortality. Keane et al. reported that patients experiencing IDH within the first hour of CRRT had a hazard ratio (HR) for all-cause mortality of 1.4 (95%$CI$:1.2–1.7) and a HR for cardiovascular mortality of 1.6 (95%$CI$:1.2–2.2) [10]. IDH leads to inadequate organ perfusion and exacerbates multiple organ dysfunction. Decreased cardiac output during hypotensive episodes leads to renal hypoperfusion, exacerbating renal injury and reducing the likelihood of renal recovery. Moreover, myocardial hypoperfusion can lead to ischemia, myocardial stunning, and arrhythmias, increasing the risk of acute heart failure and sudden cardiac death [11]. Cerebral hypoperfusion further disrupts cerebral microcirculation [12], increasing the risk of ischemic stroke and neurological impairment, which can further worsen the patient's clinical condition and increase the risk of mortality. Given these serious implications, early identification of IDH during CRRT is of paramount importance.

Research on IDH during CRRT is still constrained, predominantly comprising retrospective studies. These studies are often subject to information and selection biases, and their findings may differ according to diagnostic criteria for IDH [6,7,13–15]. The occurrence of IDH in critically ill patients is affected by a combination of factors, and previous studies have not accounted for the covariance among those factors. This study analyzed the incidence of IDH in critically ill

patients within the first hour of CRRT. Least absolute shrinkage and selection operator (LASSO) regression was used to screen the potential influencing factors and reduce collinearity, and then explored the independent variables associated with hypotension. This study was expected to provide valuable information to mitigate the occurrence of IDH during the commencement of CRRT.

## Methods

### Study design and population

This prospective observational study involved a cohort of adult patients (≥18 years old) undergoing CRRT, admitted to the central intensive care unit (ICU) of the Third Affiliated Hospital of Soochow University from January 17, 2024, to December 25, 2024. The Institutional Ethics Committee approved the project (No. F-IRB-SOP-00710), and all patients or their relatives signed the informed consent form.

### Inclusion and exclusion criteria

The inclusion criteria for the study were as follows:(1) patients aged ≥18 years old; (2) APACHE II score ≥15 at the day of ICU admission; (3) arterial cannulation for real-time hemodynamic monitoring;

The study's exclusion criteria included: (1) central nervous system disorders; (2) the use of extracorporeal membrane oxygenation (ECMO), ventricular assist devices, or pacemakers; and (3) patients with a mean arterial pressure (MAP) < 65 mmHg before the start of CRRT.

The study's dropout criteria included: (1) mortality during the observation period (the first hour of CRRT); (2) cessation of CRRT for non-hemodynamic reasons; (3) alteration in ventilation or sedation during the observation period; and (4) premature intervention before meeting diagnostic criteria for hypotension.

### Definition of hypotension within the first hour of CRRT

There is no definitive definition of IDH. The prevalent definition of IDH, as articulated by the Kidney Disease Outcomes Quality Initiative (KDOQI) of the National Kidney Foundation (USA), is characterized by a decrease in systolic blood pressure (SBP) of ≥20 mmHg or MAP of ≥10 mmHg, accompanied by symptoms such as abdominal discomfort, yawning, sighing, nausea, vomiting, muscle cramps, restlessness, dizziness, and anxiety necessitating intervention [16]. This definition is not applicable to critically ill patients, as they cannot articulate pertinent symptoms due to impaired mental status or mechanical ventilation. Moreover, significant discrepancies exist in SBP measured at various arteria cannulation sites, while MAP exhibits lesser variations [17]. It has been demonstrated that a MAP below 65 mmHg serves as the threshold for the onset of new acute myocardial and renal injury [18]. While we recognize that relative hypotension-defined as a significant drop from baseline MAP- may have clinical implications, particularly in patients with chronic hypertension, there is currently no consensus on its threshold in the CRRT setting, and supporting evidence remains limited [19]. Therefore, we did not include relative hypotension in our definition and defined IDH as a new MAP<65 mmHg lasting more than one minute during the first hour of CRRT, consistent with the definition proposed by Chazot et al. [15].

### Experimental methods

**Instruments and equipment.** The instruments and equipment used in this study included: (1) Hemodialysis machine and tubing: Gambro Hemodialysis Machine Prismaflex with corresponding tubing (2) Filters: Gambro M100 or M150; (3) Monitors: Mindray Benevision N15; (4) Arterial pressure transducer: icumedical,01C-42584–25

**Procedure for CRRT operation.** The CRRT operation adhered to the Continuous Renal Replacement Therapy Nursing Group Standard established by the Chinese Nursing Association in 2023 [20]. Operations were performed by nurses qualified for CRRT. The parameters were set according to medical advice.

**Data collection.** We developed the data collection form by reviewing pertinent literature on the determinants of CRRT-associated hypotension in databases including China National Knowledge Infrastructure (CNKI), Wanfang, and PubMed et al., combined with clinical practice and discussion during the expert meeting, to identify the potential influencing factors for hypotension in critically ill patients undergoing CRRT [6–9,13,15,21,22]. The form encompassed (1) clinical characteristics: sex, age, body mass index (BMI), Acute Physiology, and Chronic Health Evaluation II (APACHE II) at ICU admission day, duration of ICU stay at CRRT initiation, age-adjusted Charlson Comorbidity Index (aCCI), comorbidities (hypertension, diabetes, heart failure), and APACHE II and Sequential Organ Failure Assessment (SOFA) on the day of CRRT; (2) treatments and interventions: use of vasopressors, vasoactive inotropic score (VIS), use of colloid solutions, and mechanical ventilation. (3) laboratory results: lactate, pH, $PaCO_2$, white blood cell count (WBC), platelet count (PLT), hemoglobin (Hb), albumin, C-reactive protein (CRP), blood urea nitrogen (BUN), serum natrium (Na), serum kalium (K), serum calcium (Ca), serum phosphate (P), N-terminal pro-B-type natriuretic peptide (NT-proBNP), international normalized ratio (INR), myoglobin, and troponin I; (4) CRRT parameters: CRRT modality, initial blood flow rate, therapeutic blood flow rate, replacement fluid rate, ultrafiltration rate;(5) baseline vital signs: temperature, heart rate (HR), SBP, diastolic blood pressure (DBP), MAP, and respiratory rate (RR).

Before data collection, researchers received standardized training to ensure consistency in the data collection process. Clinical characteristics and laboratory data were extracted from the hospital's medical records system. Laboratory results were chosen from the most recent tests performed within 24 hours before the start of CRRT. If no such data were available, the values were treated as missing. Treatments and interventions were assessed and recorded before the CRRT initiation; CRRT parameters were recorded following parameter adjustments.

Before initiating CRRT, patients were maintained in a tranquil state and positioned in a semi-recumbent posture. Subsequently, arterial blood pressure was calibrated and set to zero. Baseline vital signs were documented when the waveform exhibited stability. Vital signs were continuously observed by a designated researcher until the occurrence of IDH or the end of the observation period. Non-urgent operations were mostly circumvented to reduce interference. All data were examined and documented by two researchers. The IDH and non-IDH groups were categorized based on the occurrence of IDH throughout the observation period.

## Statistical analysis

The proportion of missing data ranged from 0.0% to 18.3%, with the highest rates observed in P (18.3%) and NT-proBNP (18.1%). Little's MCAR test was conducted to assess the missing data mechanism, and the result showed that the data were not missing completely at random ($P < 0.001$). Given that the missingness was likely related to observed patient characteristics rather than the unobserved values themselves, it was considered reasonable to assume that the data were missing at random. Under this assumption, missing data were handled using multiple imputation via chained equations (MICE). All variables intended for inclusion in the final analysis were incorporated into the imputation model. Five imputed datasets were generated and subsequently analyzed, with parameter estimates combined using Rubin's rules. Continuous variables were summarized as mean and standard deviation (Mean±SD) or median and interquartile range [$M$ ($Q_1$, $Q_3$)], contingent upon the normality of the data distribution, while categorical variables were reported as counts and percentages [$n$ (%)]. The independent $t$-test or Wilcoxon rank-sum test was used for continuous variables, and the Chi-squared test or *Fisher's* exact test was applied for categorical variables. Variables exhibiting statistically significant variations between the two groups ($P < 0.05$) were subsequently analyzed using LASSO regression to ascertain probable important factors. LASSO regression systematically identifies a subset of variables by applying an L1-norm penalty, hence mitigating any collinearity. The tuning parameter lambda ($\lambda$), which dictates the extent of shrinkage, was chosen by 10-fold cross-validation. Variables were preserved according to the coefficients associated with the minimum lambda ($\lambda$). Multivariate logistic regression with stepwise backward selection was conducted to identify independent risk factors for IDH during the first hour of CRRT. We used Shapley Additive Explanations (SHAP) values to illustrate the importance of each variable

in the model. A two-tailed *P*-value of <0.05 was considered statistically significant. All statistical analyses were performed using *R* Studio (version 4.2.3).

## Results

### The characteristics of patients

A total of 496 patients of CRRT were recorded in this study, with 31 excluded and 30 dropped, leaving 435 for the final analysis (Fig 1). Among these, 166 patients developed IDH, representing a prevalence rate of 38.2%. The median age of patients was 71 (58,78), and 308 patients (70.64%) were male. The median APACHE II score on the day of ICU admission was 28 (21,33). There were 300 (68.97%) patients with hypertension and 180 (41.28%) with diabetes.

### Comparison between IDH and non-IDH group

Comparison between the two groups revealed that patients in the IDH group presented with more severe clinical conditions and experienced significantly prolonged ICU stays before the start of CRRT compared to those in the non-IDH group. Additionally, the IDH group exhibited lower levels of Hb, PLT, INR, baseline blood pressure and RR, along with higher levels of CRP and myoglobin. The proportions of males (79.52% vs. 65.06%, *P*=0.001), the use of vasopressors (78.92% vs. 64.68%, *P*=0.002), and mechanical ventilation (80.12% vs. 69.89%, *P*=0.018) were significantly higher in the IDH group. Conversely, the use of colloidal solutions was lower in the IDH group. Details are provided in Table 1.

### Selection of influencing factors

We performed LASSO regression analysis based on the 21 factors that showed significant differences between the two groups, including age, sex, APACHE II at ICU admission day, duration of ICU stay at CRRT initiation, APACHE II and SOFA on the day of CRRT initiation, VIS, the use of vasopressors, the use of colloidal solutions, mechanical ventilation, Hb, PLT, CRP, P, INR, myoglobin, ultrafiltration rate, RR, SBP, DBP, and MAP. Fig 2 illustrates the coefficient trajectories in the LASSO regression model and Fig 3 displays the cross-validation error. The coefficients are presented in S1 Table. The results indicated that, when the tuning parameter λ was set to 0.036 (one standard error of the minimum λ), 19 variables

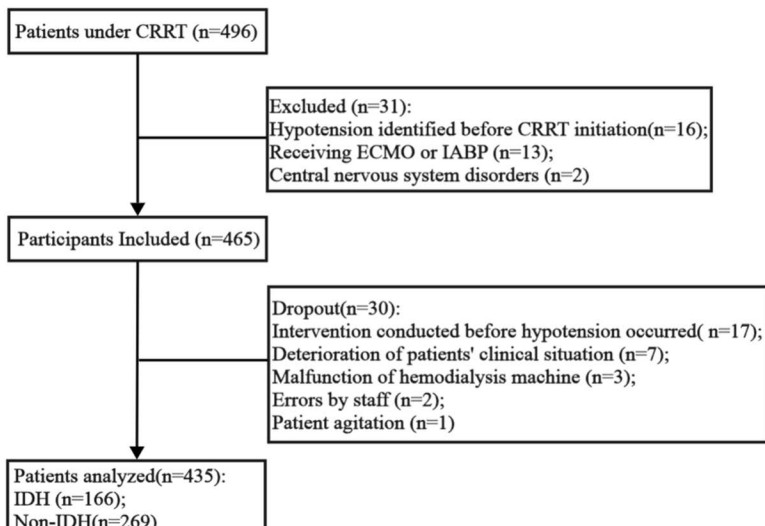

**Fig 1. Flow Chart of Participant Inclusion and Exclusion.** *CRRT, continuous renal replacement therapy; ECMO, extracorporeal membrane oxygenation; IABP, intra-aortic balloon pump.

**Table 1. Comparison between IDH and non-IDH group.**

| Variables | Total (*n*=435) | Non-IDH (*n*=269) | IDH (*n*=166) | *P*-value |
|---|---|---|---|---|
| **Clinical characteristics** | | | | |
| Age, years [*M* (Q$_1$, Q$_3$)] | 71 (58, 78) | 74 (63, 81.75) | 69 (56, 77) | **<0.001**[c] |
| Sex [*n* (%)] | | | | **0.001**[b] |
| male | 307 (70.57) | 175 (65.06) | 132 (79.52) | |
| female | 128 (29.43) | 94 (34.94) | 34 (20.48) | |
| BMI [Mean±SD] | 23.60±4.02 | 23.89±4.11 | 23.12±3.83 | 0.050[a] |
| Comorbidities | | | | |
| Hypertension [*n* (%)] | | | | 0.918[b] |
| yes | 300 (68.97) | 186 (69.14) | 114 (68.67) | |
| no | 135 (31.03) | 83 (30.86) | 52 (31.33) | |
| Diabetes [*n* (%)] | | | | 0.388[b] |
| yes | 180 (41.38) | 107 (39.78) | 73 (43.98) | |
| no | 255 (58.62) | 162 (60.22) | 93 (56.02) | |
| Heart failure [*n* (%)] | | | | 0.948[b] |
| yes | 31 (7.13) | 19 (7.06) | 12 (7.23) | |
| no | 404 (92.87) | 250 (92.94) | 154 (92.77) | |
| aCCI [Mean±SD] | 5.06±2.92 | 4.94±2.95 | 5.25±2.87 | 0.279[a] |
| APACHEIIat the day of ICU admission [*M* (Q$_1$, Q$_3$)] | 28 (21, 33) | 27 (21, 32) | 29 (23, 35.75) | **0.001**[c] |
| Duration of ICU stay at CRRT initiation, day [*M* (Q$_1$, Q$_3$)] | 4 (1, 9) | 3 (0, 8) | 5 (2, 11) | **<0.001**[c] |
| APACHE II on the day of CRRT initiation [Mean±SD] | 23.42±6.69 | 22.73±6.95 | 24.54±6.09 | **0.006**[a] |
| SOFA on the day of CRRT initiation [Mean±SD] | 11.47±4.16 | 10.81±4.26 | 12.54±3.76 | **<0.001**[a] |
| **Treatments and interventions** | | | | |
| VIS [M (Q$_1$, Q$_3$)] | 10 (0, 30) | 5 (0, 25) | 10 (2, 30) | **0.012**[c] |
| Use of vasopressors [*n* (%)] | | | | **0.002**[b] |
| yes | 305 (70.11) | 174 (64.68) | 131 (78.92) | |
| no | 130 (29.89) | 95 (35.32) | 35 (21.08) | |
| Use of colloidal solutions [*n* (%)] | | | | **0.003**[b] |
| yes | 84 (19.31) | 64 (23.79) | 20 (12.05) | |
| no | 351 (80.69) | 205 (76.21) | 146 (87.95) | |
| Mechanical ventilation [*n* (%)] | | | | **0.018**[b] |
| yes | 321 (73.79) | 188 (69.89) | 133 (80.12) | |
| no | 114 (26.21) | 81 (30.11) | 33 (19.88) | |
| **Laboratory results** | | | | |
| Lactate, mmol/L [*M* (Q$_1$, Q$_3$)] | 1.50 (1.10, 2.60) | 1.40 (1.10, 2.30) | 1.70 (1.10, 3.00) | 0.113[c] |
| pH [*M* (Q$_1$, Q$_3$)] | 7.40 (7.32, 7.46) | 7.40 (7.32, 7.46) | 7.40 (7.33, 7.45) | 0.762[c] |
| PaCO$_2$, mmHg [Mean±SD] | 41.77±10.73 | 41.26±10.53 | 42.59±11.04 | 0.211[a] |
| WBC, 10$^9$/L [*M* (Q$_1$, Q$_3$)] | 11.49 (8.29, 16.56) | 11.49 (8.43, 16.56) | 11.49 (8.01, 16.48) | 0.620[c] |
| Hb, g/L [*M* (Q$_1$, Q$_3$)] | 82 (72, 99) | 84 (72, 104) | 79 (71, 93) | **0.032**[c] |
| PLT, 10$^9$/L [*M* (Q$_1$, Q$_3$)] | 122 (76, 184) | 132 (80, 195) | 104 (69, 162) | **0.002**[c] |
| CRP, mg/L [*M* (Q$_1$, Q$_3$)] | 105.20 (47.40, 150.60) | 94.30 (40.10, 144.40) | 114.20 (58.72, 161.95) | **0.009**[c] |
| Serum natrium, mmol/L [Mean ± SD] | 139.22±6.52 | 139.27±6.47 | 139.13±6.62 | 0.836[a] |
| Serum kalium, mmol/L [Mean ± SD] | 4.44±0.67 | 4.42±0.68 | 4.46±0.64 | 0.555[a] |
| Serum calcium, mmol/L [*M* (Q$_1$, Q$_3$)] | 1.21 (1.07, 2.14) | 1.22 (1.07, 2.08) | 1.20 (1.08, 2.17) | 0.837[c] |

*(Continued)*

**Table 1.** (Continued)

| Variables | Total (*n* = 435) | Non-IDH (*n* = 269) | IDH (*n* = 166) | *P*-value |
|---|---|---|---|---|
| Serum phosphate, mmol/L [*M* (Q₁, Q₃)] | 0.88 (0.57, 1.50) | 0.99 (0.59, 1.52) | 0.79 (0.53, 1.36) | **0.021** [c] |
| Creatinine, μmmol/L [*M* (Q₁, Q₃)] | 155.00 (96.50, 281.40) | 157.70(99.00,319.00) | 148.75 (92.25, 212.75) | 0.079 [c] |
| BUN, mmol/L [*M* (Q₁, Q₃)] | 10.98 (6.80, 19.52) | 11.03 (6.57, 20.74) | 10.47 (7.31, 16.46) | 0.447 [c] |
| Albumin, g/L [Mean ± SD] | 31.35 ± 5.11 | 31.21 ± 5.22 | 31.59 ± 4.93 | 0.452 [a] |
| NT-proBNP, ng/L[*M* (Q₁, Q₃)] | 5780 (1810, 18750) | 5600(1680, 24900) | 5840(2042, 13750) | 0.807 [c] |
| INR [*M* (Q₁, Q₃)] | 1.24 (1.13, 1.47) | 1.22 (1.13, 1.41) | 1.28 (1.15, 1.50) | **0.034** [c] |
| Myoglobin, ng/ml[*M*(Q₁, Q₃)] | 294.70 (128.75, 838.30) | 229.90(122.10,762.80) | 441.40(157.25,1257.95) | **<0.001** [c] |
| Troponin I, ng/ml[*M* (Q₁, Q₃)] | 0.16 (0.04, 1.35) | 0.13 (0.04, 1.13) | 0.18 (0.04, 1.43) | 0.277 [c] |
| **CRRT parameters** | | | | |
| CRRT modality [*n* (%)] | | | | 0.090 [b] |
| CVVH | 399 (91.72) | 242 (89.96) | 157 (94.58) | |
| CVVHD | 36 (8.28) | 27 (10.04) | 9 (5.42) | |
| Initial blood flow rate, ml/min [*n* (%)] | | | | 0.057 [b] |
| 0-50 | 11 (2.53) | 3 (1.12) | 8 (4.82) | |
| 51-100 | 398 (91.49) | 250 (92.94) | 148 (89.16) | |
| >100 | 26 (5.98) | 16 (5.95) | 10 (6.02) | |
| Therapeutic blood flow rate, ml/min [*n* (%)] | | | | 1.000 [d] |
| 100-120 | 3 (0.69) | 2 (0.74) | 1 (0.60) | |
| 121-150 | 18 (4.14) | 11 (4.09) | 7 (4.22) | |
| 151-180 | 413 (94.94) | 255 (94.80) | 158 (95.18) | |
| >180 | 1 (0.23) | 1 (0.37) | 0 (0.00) | |
| Replacement fluid rate, ml/h [*n* (%)] | | | | 0.209 [d] |
| 1000-1500 | 4 (0.92) | 4 (1.49) | 0 (0.00) | |
| 1501-2000 | 73 (16.78) | 50 (18.59) | 23 (13.86) | |
| 2001-2500 | 334 (76.78) | 202 (75.09) | 132 (79.52) | |
| >2500 | 24 (5.52) | 13 (4.83) | 11 (6.63) | |
| Ultrafiltration rate, ml/h [*n* (%)] | | | | **0.043** [d] |
| 0-100 | 311 (71.49) | 182 (67.66) | 129 (77.71) | |
| 101-200 | 88 (20.23) | 62 (23.05) | 26 (15.66) | |
| 201-300 | 30 (6.90) | 19 (7.06) | 11 (6.63) | |
| >300 | 6 (1.38) | 6 (2.23) | 0 (0.00) | |
| **Baseline vital signs** | | | | |
| Temperature, °C [Mean ± SD] | 37.31 ± 0.96 | 37.36 ± 0.92 | 37.24 ± 1.02 | 0.207 [a] |
| RR, rpm [Mean ± SD] | 18.26 ± 4.26 | 17.91 ± 4.22 | 18.83 ± 4.29 | **0.030** [a] |
| HR, bpm [Mean ± SD] | 96.14 ± 22.29 | 95.44 ± 22.13 | 97.28 ± 22.56 | 0.405 [a] |
| SBP, mmHg [Mean ± SD] | 143.46 ± 23.07 | 148.36 ± 21.46 | 135.53 ± 23.44 | **<0.001** [a] |
| DBP, mmHg [Mean ± SD] | 64.74 ± 14.44 | 68.30 ± 13.36 | 58.99 ± 14.32 | **<0.001** [a] |
| MAP, mmHg [Mean ± SD] | 90.59 ± 15.15 | 94.45 ± 13.87 | 84.34 ± 15.09 | **<0.001** [a] |

*[a] *t*-test; [b] chi-square test; [c] Wilcoxon rank sum test; [d] Fisher's exact test; IDH, intra-dialytic hypotension; BMI, body mass index; aCCI, age-adjusted Charlson Comorbidity Index; APACHE II, Acute Physiology, and Chronic Health Evaluation II; SOFA, Sequential Organ Failure Assessment; VIS, vaso-active inotropic score; WBC, white blood cell count; Hb, hemoglobin; PLT, platelet count; CRP, C-reactive protein; BUN, blood urea nitrogen; NT-proBNP, N-terminal pro-B-type natriuretic peptide; INR, international normalized ratio; CRRT, continuous renal replacement therapy; CVVH, Continuous Veno-Venous Hemofiltration; CVVHD, Continuous Veno-Venous Hemodiafiltration; RR, respiratory rate; HR, heart rate; SBP, systolic blood pressure; DBP, diastolic blood pressure; MAP, mean arterial pressure.

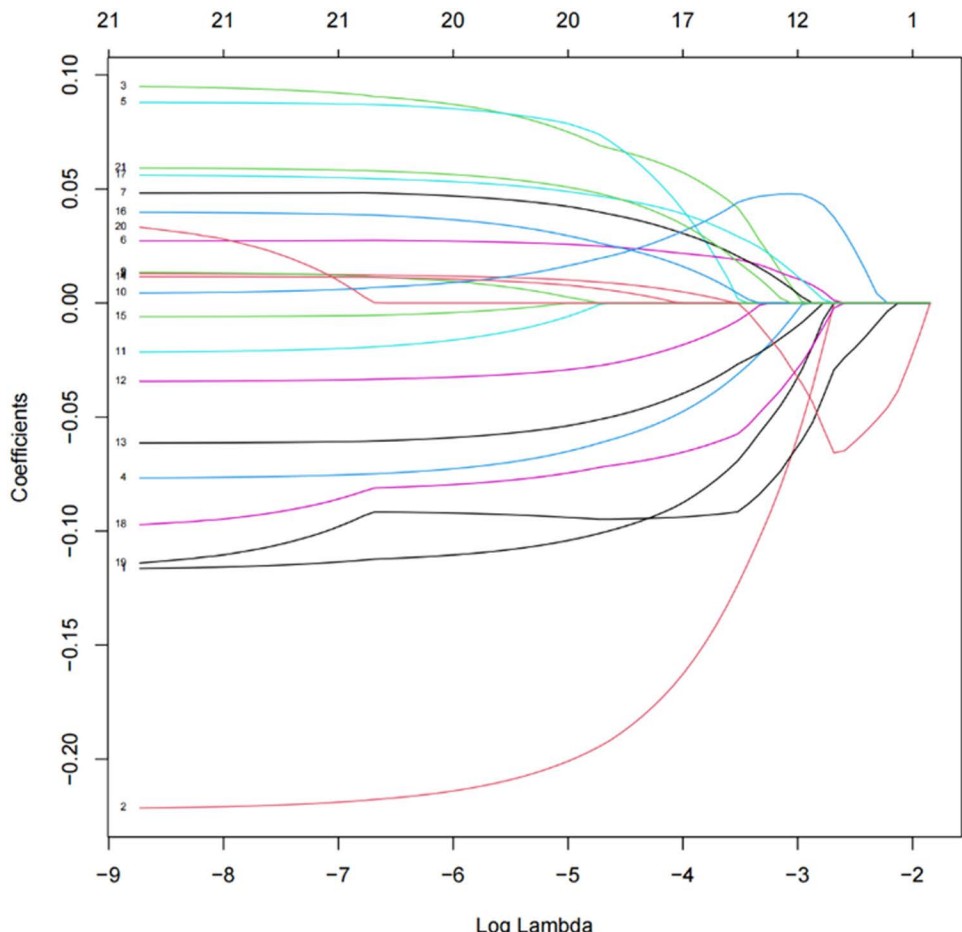

**Fig 2. LASSO Regression Coefficient Path Plot.**

were selected in the model, except for APACHE II on the day of CRRT initiation and baseline MAP. Although baseline MAP was excluded, we believe that SBP and DBP have a strong correlation. Given the definition of hypotension in this study, we propose that baseline MAP may serve as a more reliable predictor. Therefore, we decided to exclude SBP and DBP and include MAP in the model.

We compared the performance of models including either SBP or MAP. The model including MAP showed a lower Akaike Information Criterion (AIC) of 466.76, which is better than the model with SBP (AIC: 472.52). The result of the multivariate logistic regression model with MAP included is shown in Table 2. Female ($OR$=0.53, 95%$CI$:0.3–0.89), the use of colloidal solutions ($OR$=0.23, 95%$CI$:0.12–0.46), PLT ($OR$=0.99, 95%$CI$:0.99–0.99), and baseline MAP ($OR$=0.95, 95%$CI$:0.93–0.96) were recognized as protective factors against IDH. Older age ($OR$=1.02, 95%$CI$:1.01–1.04), mechanical ventilation ($OR$=2.59, 95%$CI$:1.28–5.23), ultrafiltration rates of 101–200 ml/h ($OR$=2.04, 95%$CI$:1.11–3.75), INR ($OR$=1.78, 95%$CI$:1.02–3.09), and myoglobin levels ($OR$=1.01, 95%$CI$:1.01–1.01) were identified as significant risk factors for IDH.

SHAP values were computed to evaluate the importance of each predictor in the model. A bar plot (Fig 4) was generated to visualize the importance of the predictors, with features ranked according to their average impact on the model's output. The results indicate that baseline MAP and mechanical ventilation are the two most important predictors.

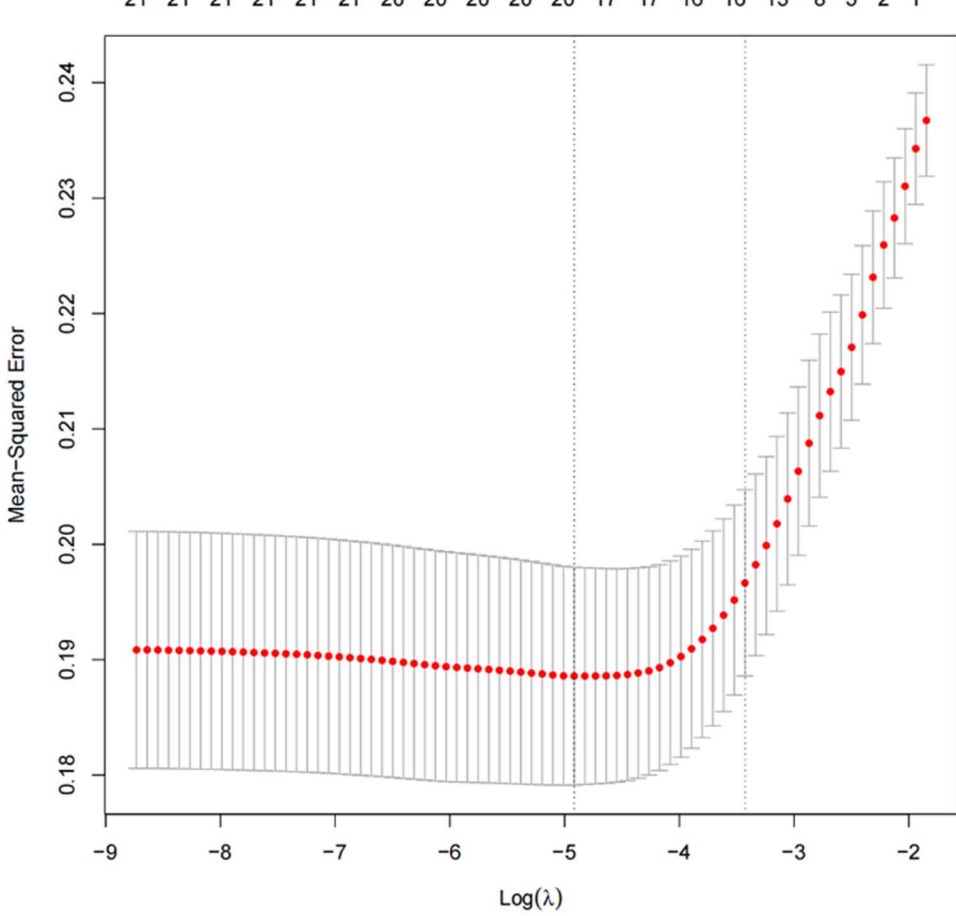

**Fig 3. Tuning Parameter (λ) Selection Cross-validation Error Curve.**

## Discussion

CRRT is widely used in critically ill patients owing to its superior hemodynamic stability compared to other dialysis modalities. During CRRT, patients typically lose approximately 150–200 ml of circulating blood volume [23]. This rapid decrease in blood volume may not be compensated for critically ill patients, which causes hemodynamic instability. Hypotension may occur even with interventions such as bidirectional blood withdrawal. There is no standardized definition of hypotension during CRRT in critically ill patients. Depending on different definitions, previous studies have reported the incidence of hypotension within the first hour of CRRT ranging from 8.4% to 64.6% [7–9,24] and in our study was 38.2%. Therefore, early identification of IDH risk factors and close hemodynamic monitoring at the start of CRRT are essential for timely prevention and management.

Our findings align with previous studies [6,8,21,25,26], which demonstrated an independent correlation between age and the occurrence of hypotension during CRRT. Jing Li et al. reported that patients aged ≥60 years old have a higher risk of hypotension during CRRT ($OR$=6.125, 95%$CI$: 2.51–14.97) [6]. This is primarily associated with the physiological decline observed in the elderly, characterized by changes in myocardial function, including diminished myocardial contractility and left ventricular diastolic dysfunction. Simultaneously, diminished arterial endothelial function leading to decreased vascular compliance, along with autonomic dysfunction, impairs the rapid compensatory mechanisms for hemodynamic

**Table 2. Multivariate logistic regression analysis of IDH.**

| Variables | β | S. E | Z | P-value | OR (95%CI) |
|---|---|---|---|---|---|
| Intercept | −0.98 | 1.39 | −0.71 | 0.478 | 0.37 (0.02–5.66) |
| Age, year | 0.02 | 0.01 | 2.73 | **0.006** | 1.02 (1.01–1.04) |
| Sex | | | | | |
| male | | | | | 1.00 (Reference) |
| female | −0.65 | 0.28 | −2.36 | **0.018** | 0.53 (0.3–0.89) |
| Mechanical ventilation | | | | | |
| no | | | | | 1.00 (Reference) |
| yes | 0.95 | 0.35 | 2.68 | **0.007** | 2.59 (1.28–5.23) |
| Use of Colloidal solutions | | | | | |
| no | | | | | 1.00 (Reference) |
| yes | −1.45 | 0.36 | −4.06 | **<0.001** | 0.23 (0.12–0.46) |
| Use of vasopressors | | | | | |
| no | | | | | 1.00 (Reference) |
| yes | 0.49 | 0.33 | 1.48 | 0.138 | 1.64 (0.85–3.14) |
| Ultrafiltration rate, ml/h | | | | | |
| 0-100 | | | | | 1.00 (Reference) |
| 101-200 | 0.71 | 0.31 | 2.30 | **0.022** | 2.04 (1.11–3.75) |
| 201-300 | 0.10 | 0.53 | 0.19 | 0.847 | 1.11 (0.39–3.1) |
| >300 | −14.56 | 860.82 | −0.02 | 0.987 | 0.00 (0.00–Inf) |
| Hb, g/L | −0.01 | 0.01 | −1.74 | 0.082 | 0.99 (0.98–1.00) |
| PLT, 10$^9$/L | −0.01 | 0.00 | −3.14 | **0.002** | 0.99 (0.99–0.99) |
| INR | 0.58 | 0.28 | 2.04 | **0.041** | 1.78 (1.02–3.09) |
| Myoglobin, ng/ml | 0.01 | 0.00 | 3.07 | **0.002** | 1.01 (1.01–1.01) |
| MAP, mmHg | −0.05 | 0.01 | −5.93 | **<0.001** | 0.95 (0.93–0.96) |
| RR, rpm | 0.09 | 0.03 | 2.95 | **0.003** | 1.10 (1.03–1.16) |

\* *S. E*, standard error; *OR*, odds ratio; *CI*, confidence interval; Hb, hemoglobin; PLT, platelet count; INR, international normalized ratio; MAP, mean arterial pressure; RR, respiratory rate.

alterations in elderly patients, significantly reducing the ability to endure fluctuations in blood volume [27]. In addition, elderly patients commonly have multiple chronic comorbidities, which may increases the risk of hypotension during CRRT [28]. In our study, among patients aged ≥60 years old, the proportions with hypertension, diabetes, and heart failure were 74.76%, 44.41%, and 44.09%, respectively, significantly higher than those among patients aged <60 years old (53.66%, 33.61%, and 26.83%, respectively).

In this study, female was considered as a protective factor (*OR*=0.53, 95%*CI*: 0.3–0.89), which is consistent with the findings of Juan C et.al (*OR*=0.43, 95%*CI*: 0.3–0.89) [29]. We conducted a stratified analysis by sex to further explore differences in clinical characteristics such as age and comorbidities. The results showed no statistically significant differences between male and female patients in terms of these variables, and the overall severity of illness was also comparable, as reflected by similar median SOFA scores (12 vs. 11) and APACHE II scores (27 vs. 28). The proportion of female patients admitted to the ICU was relatively low (20.48%). These findings are consistent with previous studies reporting sex-related disparities in ICU admissions [30]. Further analysis indicated that male patients exhibited poorer parameters related to respiratory function. This may be associated with a higher prevalence of smoking among male individuals. According to a report published by the Chinese Center for Disease Control and Prevention (China CDC) in 2020, the smoking rate among males aged 15 years and older was 50.5%, substantially higher than that among females

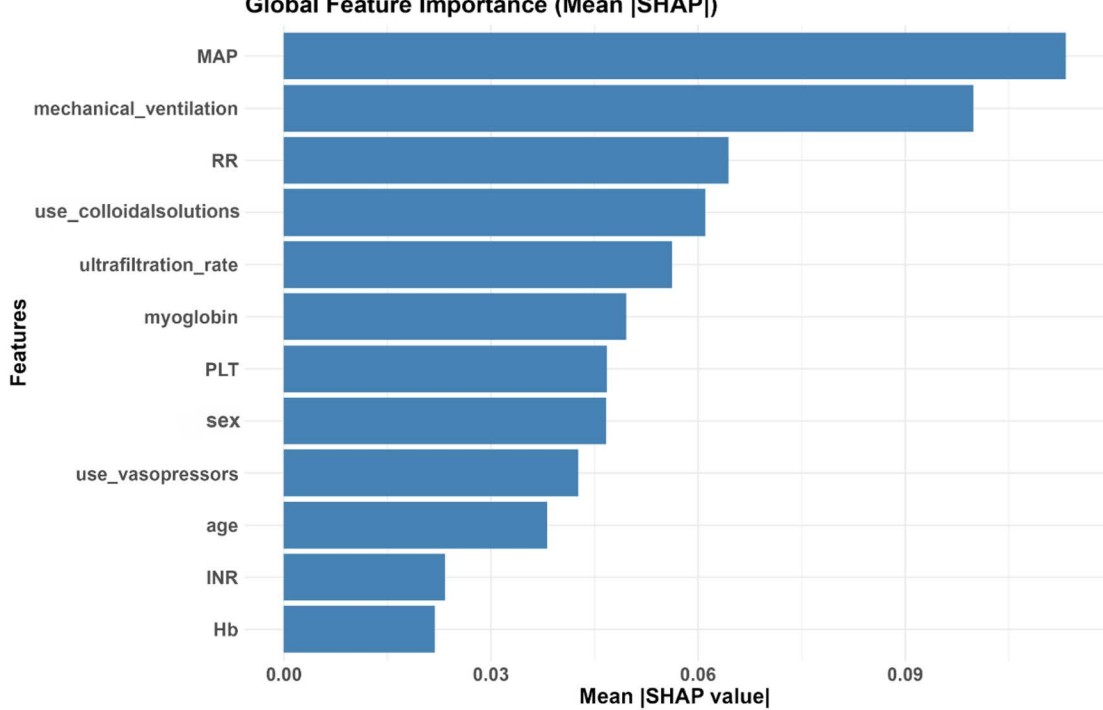

**Fig 4. Importance of Predictors Based on SHAP Value.** *MAP, mean arterial pressure; RR, respiratory rate; PLT, platelet count; INR, international normalized ratio; Hb, hemoglobin.

(approximately 2.1%) [31]. And the proportion of male patients who received mechanical ventilation was significantly higher than that of female patients (76.42% vs. 66.41%). This difference may partially explain the observed protective association of female sex in the development of hypotension.

Mechanical ventilation is one of the most important independent risk factors for hypotension during the first hour of CRRT in critically ill patients, aligning with the findings of Kelly et al [13]. In critically ill patients, mechanical ventilation implies severe respiratory failure. According to the mechanism of cardiopulmonary interaction, mechanical ventilation can increase intrathoracic negative pressure and decrease return blood volume, which may lead to severe hemo-dynamic consequences in certain clinical settings, such as chronic obstructive pulmonary disease (COPD), acute respiratory distress syndrome (ARDS), and acute left heart failure [32]. This study also found that the use of colloi-dal solutions (e.g., albumin, low molecular weight dextran, etc.) reduced the incidence of hypotension, corroborating earlier studies [6,8,21,24]. Due to higher molecular weight and stronger colloidal osmotic pressure, colloidal solutions are effective in sustaining intravascular volume, and reducing the occurrence of hypotension [33]. For patients with circulatory failure, selecting the suitable colloidal solutions based on the specific clinical situation before CRRT initia-tion may help reduce the risk of hypotension.

We found PLT is a protective factor against IDH during CRRT. Thrombocytopenia is frequently observed in critically ill patients and occurs in conditions such as septic shock, multiple organ dysfunction syndrome, and disseminated intra-vascular coagulation. The underlying mechanisms including increased platelet consumption, splenic sequestration, and accelerated clearance mediated by systemic inflammation et al. To further explore this association, patients were stratified into three groups based on platelet count: $< 100 \times 10^9$/L, $100-300 \times 10^9$/L, and $>300 \times 10^9$/L. The results showed that the

use of vasopressors was significantly higher in the thrombocytopenia group (<100 × 10⁹/L), with a rate of 83.33%, compared to 59.83% and 67.86% in the other two groups. Additionally, the mean SOFA score in the thrombocytopenia group was 13.77, higher than those in the other two groups (9.88 and 10.29, respectively).These fingdings suggest that reduced PLT may relect more severe illness and greater hemodynamic compromise.Kang et al. found INR as a risk factor for hypotension in their machine learning model [24], aligning with this study's findings, although it was ranked lower in terms of importance. Myoglobin was identified for the first time as a risk factor for hypotension within the first hour of CRRT in critically ill patients in this study. This may be related to the high prevalence of infection in this population and the potential impairment of cardiomyocytes caused by inflammatory mediators. In our cohort, the median CRP level was higher in patients with hypotension compared to thouse without (114.2 vs. 94.3 mg/L), supporting the presence of a heightened inflammatory response. A global epidemiological analysis of ICU infection indicated that around 54% of ICU patients had suspected or confirmed infections [34]. Myoglobin is known not only for its direct nephrotoxic effects, but also for contributing to endothelial injury and microvascular dysfunction, which may exacerbates hemodynamic instability and increase the risk of hypotension during CRRT.

Several studies have confirmed that ultrafiltration rate is an independent risk factor for hypotension during CRRT in critically ill patients [6–8,21], with rates ≥100 ml/h significantly increasing the risk of hypotension [6,21]. The results of this study show that the risk of hypotension was significantly increased when the ultrafiltration rate was 101–200 ml/h compared to the ultrafiltration rate of 0–100 ml/h. However, no significant difference was observed for ultrafiltration rates exceeding 200 ml/h between the two group. This unexpected result may be explained by the limited number of patients in this subgroup, which likely reduced the statistical power to detect a difference. Additionally, it is possible that clinicians often adopt a more conservative ultrafiltration strategy in order to reduce the risk of IDH, thus attenuating the observed impact of higher ultrafiltration rates. High ultrafiltration rates imply that more fluid is removed, and critically ill patients may not be able to compensate for the rapid decline in blood volume. To address this, Sun et al. proposed a three-stage volume management strategy that provides important guidance for individualized ultrafiltration, suggesting that the net ultrafiltration rate for CRRT should start at a low dose, with an initial net ultrafiltration rate of 80 (49,111) ml/h for hemodynamically unstable patients [35], to reduce the risk of hypotension and provide a more balanced and individualized ultrafiltration plan for patients.

In this study, the use of vasopressors was not found to be an independent risk factor for hypotension during the first hour of CRRT, a variable that may have a strong correlation with baseline blood pressure (MAP). It still had a significant impact on the occurrence of hypotension. Previous studies have shown that patients receiving vasopressors were at higher risk of hypotension [6–8]. One-third of ICU patients experience circulatory failure and require vasopressors for main organ perfusion [36], which may mask the true hemodynamic instability of the patient. The filter's operation can affect the pharmacokinetics of vasopressors. Vasopressors with high unbound fractions (e.g., epinephrine and norepinephrine) are more likely to be effectively removed by CRRT [37]. The present feature ranking analysis demonstrated that baseline MAP is the most important contributor to hypotension, which should be assessed before CRRT initiation. It is recommended that the dosage of vasopressors should be adjusted appropriately to maintain blood pressure at least 10 mmHg above the target [38].

There were several limitations of this study. As a single-center study, the sample size of this study was relatively small, and the findings may not be widely applicable. Further validation with multicenter and large samples is needed. Besides, this study did not exclude patients with a history of hypotension during CRRT, which may have a potential impact on the results. Although relative hypotension may have clinical significance, it was not included in our definition due to the lack of a standardized threshold and limited supporting evidence. Lastly, this study was an observational study, susceptible to unmeasured biases and unknown confounding factors, a relationship between hypotension and the factors could not be inferred causally. The findings of this study are considered a primary reference for clinical practice.

## Conclusion

The incidence of hypotension within the first hour of CRR in critically ill patients was 38.2%. Multiple factors affect the occurrence of hypotension. Female, PLT, and baseline MAP are protective factors against hypotension, whereas age, mechanical ventilation, ultrafiltration rate of 101–200 ml/h, INR, and myoglobin levels are risk factors for hypotension. More prospective studies are required to identify key determinants and develop targeted strategies to prevent hypotension.

## Supporting information

**S1 Table. The coefficients of LASSO regression analysis.**
(DOCX)

## Acknowledgments

The authors would like to acknowledge the help provided by clinicians and nurses in the ICU during data collection, and the guidance of data analysis provided by Mengsheng Zhao from Nanjing Medical University.

## Author contributions

**Conceptualization:** Jingjuan Xu, Mengdie Xue, Suying Lu, Chenglin Zhao.

**Data curation:** Mengdie Xue, Jing Yang, Zhiyu Mao.

**Formal analysis:** Zheyao Zhang, Zhiyu Mao.

**Investigation:** Mengdie Xue, Suying Lu, Chenglin Zhao, Jing Yang.

**Methodology:** Jingjuan Xu, Mengdie Xue, Suying Lu, Chenglin Zhao.

**Project administration:** Mengdie Xue.

**Supervision:** Jingjuan Xu.

**Visualization:** Mengdie Xue, Zheyao Zhang.

**Writing – original draft:** Jingjuan Xu, Mengdie Xue.

**Writing – review & editing:** Jingjuan Xu, Mengdie Xue, Suying Lu, Chenglin Zhao.

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
