## [Decision Letter · Decision Letter 0]

Apr 19 2025

PONE-D-25-05284Factors Associated with Hypotension During the Initial Hour of Continuous Renal Replacement Therapy in Critically Ill Patients: A Prospective Observational StudyPLOS ONE

Dear Dr. Xu,

Thank you for submitting your manuscript to PLOS ONE. After careful consideration, we feel that it has merit but does not fully meet PLOS ONE’s publication criteria as it currently stands. Therefore, we invite you to submit a revised version of the manuscript that addresses the points raised during the review process.

We look forward to receiving your revised manuscript.

Kind regards,

Chiara Lazzeri

Academic Editor

PLOS ONE

Journal Requirements:

2. In the online submission form, you indicated that [The datasets are not publicly available due to the potential for loss of privacy as required by the research ethics committee. The data are available from the corresponding author and research ethics committee (helenclinical@163.com) on reasonable request.].

Reviewers' comments:

Reviewer's Responses to Questions

**Comments to the Author**

1. Is the manuscript technically sound, and do the data support the conclusions?

Reviewer #1: Yes

Reviewer #2: Yes

2. Has the statistical analysis been performed appropriately and rigorously? 

Reviewer #1: Yes

Reviewer #2: I Don't Know

3. Have the authors made all data underlying the findings in their manuscript fully available?

Reviewer #1: No

Reviewer #2: No

4. Is the manuscript presented in an intelligible fashion and written in standard English?

Reviewer #1: Yes

Reviewer #2: Yes

5. Review Comments to the Author

Reviewer #1: I am grateful for the chance to evaluate your manuscript. I have provided the following comments and recommendations to improve the lucidity, scientific rigor, and overall presentation of your work:

1- Some wording needs enhancement to improve clarity and to elaborate on the physiological mechanisms of hypotension's effect on mortality.

2- Use "patients with a mean arterial pressure (MAP) <65 mmHg" instead of "not applicable."

3- The phrase regarding missing data should be changed to: "Missing data were handled using multiple imputation techniques."

4- The statement should read: "A two-tailed P-value of <0.05 was considered statistically significant."

5- It would be beneficial to clarify the reasons why a filtration rate over 200 mL/h did not exhibit a discernible effect and why antihypertensive use was not considered an independent risk factor.

6- Conclusion Refinements:Change "this incidence" to "the incidence" and "within the initial hour" to "within the first hour."

7- Reference Formatting:Ensure compliance with the "Vancouver" style, standardizing author name formatting and unifying journal names with volume, issue, and page numbers.

Please focus on these points in the attached file during the revision of the manuscript to enhance clarity, consistency, and adherence to publication standards. Thank you for your efforts in improving the quality of the work.

Reviewer #2: Your use of IDH <65mmHg MAP does not take into account the recommended definitions of a drop in MAP (whilst understanding the reported factors may not all be applicable within critical care patients). Utilising only MAP <65mmgH will mean that this does not take into account relative hypotension and its effect. It is unclear here why only this was applied and not a persistent reduction in MAP of >10mmHg alongside this. This is especially important when 69% of your patients had pre-existing hypertension.

You give a good rationale for only using MAP in critical care patients.

You state that "missing data were inputed using multiple interpolation methods"- but I can't see anywhere that you discuss what data was missing. It would be good to include this for the readers to be able to fully interpret your findings as it is unclear from this statement how much missing data there was.

Inputed spelt incorrectly.

You discuss that being female was protective of IDH hypotension- it would have been good to understand some more of the characteristics of females (e.g. rates of associated co-morbidities/ age etc) and for this to be discussed more within your discussion.

Your discussion is generally well linked to previous work around IDH/ Critical care- but more work could be done on improving this to understand your results more by providing more analysis on the factors discussed (e.g. age/ platelets etc).

6. PLOS authors have the option to publish the peer review history of their article (what does this mean? ). If published, this will include your full peer review and any attached files.

**Do you want your identity to be public for this peer review?** For information about this choice, including consent withdrawal, please see our Privacy Policy .

Reviewer #1: No

Reviewer #2: No

---

## [Author Response · Author response to Decision Letter 1]

22 Apr 2025

Dear Editor and Reviewers,

We sincerely appreciate your time and effort in reviewing our manuscript [PONE-D-25-05284] entitled “Factors Associated with Hypotension During the Initial Hour of Continuous Renal Replacement Therapy in Critically Ill Patients: A Prospective Observational Study”. We are grateful for your insightful comments and suggestions, which have helped us improve the quality of our work. Below, we provide detailed responses to each comment. The modifications are turned to red in the revised manuscript.

Reviewer#1

1- Some wording needs enhancement to improve clarity and to elaborate on the physiological mechanisms of hypotension's effect on mortality.

Thank you for your valuable comments. We have carefully revised the manuscript to enhance clarity and provide a more detailed explanation of physiological mechanisms linking hypotension to mortality. Mortality is most closely associated with multiorgan hypoperfusion due to IDH. These revisions can be found in the [Introduction- Paragraph2].

2- Use "patients with a mean arterial pressure (MAP) <65 mmHg" instead of "not applicable."

Thank you for your suggestion. After verification, we noted that “not applicable” was not present in our original manuscript and thus no changes were made regarding this term.

3- The phrase regarding missing data should be changed to: "Missing data were handled using multiple imputation techniques."

Thanks for your advice. Based on your comment and a similar recommendation from another reviewer, we have removed the original sentence and revised the manuscript to provide a more comprehensive description of the missing data handling process. The revised version includes details on the extent of missing data, the type of missingness, and the multiple imputation technique used. The updated text can be found in the [Statistical analysis-Paragraph1]

4- The statement should read: "A two-tailed P-value of <0.05 was considered statistically significant."

Thanks for your advice. We have revised the statement as recommended to “A two-tailed P-value of <0.05 was considered statistically significant”. The updated text can be found in the section of [Statistical analysis]

5- It would be beneficial to clarify the reasons why a filtration rate over 200 mL/h did not exhibit a discernible effect and why antihypertensive use was not considered an independent risk factor.

Thank you for your insightful comment.

1. Regarding the lack of a significant effect associated with ultrafiltration rates above 200ml/h, we believe this may be attributed to the relatively small samples of patients receiving such high ultrafiltration rates in our cohort, which limited the statistical power. Additionally, in clinical practice, patients with unstable hemodynamics or higher risk of hypotension tend to receive more conservative fluid removal. As a result, high-risk patients are often managed with lower ultrafiltration rates, which may have attenuated the observed effect. This explanation has been incorporated into the revised [Discussion] section.

2. As for antihypertensive medication use, we did not include this variable in our analysis. We provide the following rationale. Firstly, previous studies have not demonstrated a significant association between antihypertensive use and the occurrence of intradialytic hypotension during CRRT. Nevertheless, we did collect relevant data. However, the proportion of patients receiving antihypertensive medications in our cohort was low, approximately 5.6%, which limited the feasibility of meaningful statistical analysis. In addition, it is common for healthcare providers to reduce the dosage or temporarily withhold antihypertensive medication using prior to initiating CRRT, which may further confound the assessment of their potential impact on early hypotension. For these reasons, antihypertensive use did not include in our analysis.

6- Conclusion Refinements: Change "this incidence" to "the incidence" and "within the initial hour" to "within the first hour."

Thank you for your suggestion. We have made the recommended refinements, changing “this incidence” to “the incidence” and “within the initial hour” to “within the first hour” in the section of [Conclusion]

7- Reference Formatting: Ensure compliance with the "Vancouver" style, standardizing author name formatting and unifying journal names with volume, issue, and page numbers.

Thank you for your comment. We have carefully revised the reference list to ensure full compliance with the Vancouver citation style. Author names, journal abbreviations, volume and issue numbers, as well as page ranges, have all been standardized. In addition, DOI has been added where available.

Reviewer#2:

1- The definition of IDH using only MAP <65mmHg does not account for relative hypotension and its potential impact. Given that 69% of patients had pre-existing hypertension, a persistent MAP reduction of >10mmHg should be considered, as a drop form a higher baseline may still have clinical significance. Clarifying why this criterion was not included.

Thank you for your thoughtful comment. We agree that relative hypotension, particularly in patients with pre-existing hypertension, may have important clinical implications. However, current research on this issue remains limited. Although some literature reviews have suggested a potential association between relative hypotension and organ dysfunction, high-quality evidence is still lacking, and there is no universally accepted definition of relative hypotension at present. In our study, we adopted the used definition of hypotension based on previous literature, without including relative hypotension as a criterion, in order to enhance the comparability of our results. We have added an explanation of this rationale in the section of [Definition of hypotension within the first hour of CRRT] and acknowledged this as a limitation in the manuscript and look forward to future high-quality studies that further explore the definition and clinical implications of relative hypotension.

2-You state that "missing data were inputed using multiple interpolation methods"- but I can't see anywhere that you discuss what data was missing. It would be good to include this for the readers to be able to fully interpret your findings as it is unclear from this statement how much missing data there was. “Inputed” spelt incorrectly.

Thank you for your valuable feedback.

1. We apologize for the spelling error, which has now been corrected to “imputed”.

2. In response to your comment, we have revised the manuscript to include a detailed description of the extent of missing data. We now report the range of missingness (0.0% to 18.3%) across all variables and note that the highest missing rates were observed in serum phosphate (18.3%) and N-terminal pro-B-type natriuretic peptide (18.1%). We also added that Little’s MCAR test indicated the data were not missing completely at random (P<0.001), and that multiple imputation was therefore conducted under the assumption of missing at random (MAR). This information has been added at the section of [Statistical analysis-Paragraph1].

3-You discuss that being female was protective of IDH hypotension- it would have been good to understand some more of the characteristics of females (e.g. rates of associated co-morbidities/ age etc) and for this to be discussed more within your discussion.

We appreciate your insightful comment. In response, we conducted a stratified analysis by sex to further explore differences in clinical characteristics such as age and comorbidities. The result showed no statistically significant differences between male and female patients in terms of these variables, and overall severity of illness was also comparable, as reflected by similar median SOFA (12 vs.11) and APACHE Ⅱ (27 vs. 28). These findings are in line with previous studies reporting sex-disparities in ICU admissions. However, further analysis revealed that male patients had poorer respiratory function–related laboratory parameters, which may be associated with a higher prevalence of smoking among males. According to a 2020 report by the Chinese Center for Disease Control and Prevention (China CDC), the smoking rate among males aged 15 years and older was 50.5%, substantially higher than that among females (approximately 2.1%). Additionally, the proportion of male patients requiring mechanical ventilation was significantly higher than that of female patients (76.42% vs. 66.41%). Since mechanical ventilation is one of the strongest independent risk factors for hypotension. That may partially explain the observed protective association of female sex in the development of hypotension. We have incorporated this discussion into the revised manuscript [Discussion-Paragraph 3], and the relationship between hypotension and mechanical ventilation was addressed in the [Disscussion-Paragraph4].

4-Your discussion is generally well linked to previous work around IDH/ Critical care- but more work could be done on improving this to understand your results more by providing more analysis on the factors discussed (e.g. age/ platelets etc).

Thanks for your insightful suggestion.

1. We performed a stratified analysis by age and found that patients aged ≥60 years old had higher proportions of hypertension, diabetes, and cardiac dysfunction compared to those aged <60 years. This may help explain the association between advanced age and the occurrence of IDH. The detail is presented in the revised [Discussion-Paragraph2]

2. We have expanded this discussion to include a more in-depth analysis of platelet count and its potential role in intradialytic hypotension. We stratified patients into three groups based on platelet count: <100, 100–300, and >300 ×10⁹/L. We found that patients with thrombocytopenia (<100 ×10⁹/L) had a significantly higher rate of vasopressor use (83.33%) compared to the other two groups (59.83% and 67.86%). Additionally, the mean SOFA score in the thrombocytopenia group was markedly higher (13.77 vs. 9.88 and 10.29, respectively; all p-values < 0.05), suggesting that low platelet count may be a marker of greater illness severity and circulatory compromise. Actually, thrombocytopenia is frequently observed in critically ill patients and occurs in conditions such as septic shock, multiple organ dysfunction syndrome, and disseminated intravascular coagulation. The underlying mechanisms including increased platelet consumption, splenic sequestration, and accelerated clearance mediated by systemic inflammation et al. These findings have been incorporated into the revised [Discussion -Paragraph5].

In addition, we replaced “gender” with “sex” in the whole manuscript to ensure the use of appropriate terminology in accordance with biomedical standards. We added the subtitle in section of [Results] to make it clear. We removed the original Table “the characteristics of patients” and replaced it with Table “Comparison between IDH and non-IDH group” to show the differences between the two groups. The raw data have been provided in the Supporting Information.

---

## [Editor Report · Decision Letter 1]

Factors Associated with Hypotension During the First Hour of Continuous Renal Replacement Therapy in Critically Ill Patients: A Prospective Observational Study

PONE-D-25-05284R1

Dear Dr. Xu,

We’re pleased to inform you that your manuscript has been judged scientifically suitable for publication and will be formally accepted for publication once it meets all outstanding technical requirements.

Kind regards,

Chiara Lazzeri

Academic Editor

PLOS ONE